# Model Construction of Chinese Preservice Physical Education Teachers’ Perception of Social Media: A Grounded Theory Approach

**DOI:** 10.3390/ijerph20053886

**Published:** 2023-02-22

**Authors:** Yue Xu, Zhihua Yin, Haohui Liu, Mingzhu Sun, Zhen Guo, Bo Liu

**Affiliations:** 1Department of Physical Education and Sport Sciences, University of Limerick, V94 T9PX Limerick, Ireland; 2College of Physical Education and Health, East China Normal University, Shanghai 200241, China; 3Department of Physical Education Teaching, Shanghai University of Engineering Science, Shanghai 201620, China; 4Division of Sports Science and Physical Education, Tsinghua University, Beijing 100084, China

**Keywords:** preservice physical education teacher, social media, grounded theory

## Abstract

(1) Background: Pre-service physical education teachers commonly embrace social media for multiple purposes. However, little is known about their perception of social media, which could affect the appropriate use of social media in their future professional work. This study aims to explore a theoretical model of how pre-service physical education teachers perceive social media in order to provide a basis for educators to guide their appropriate use of social media. (2) Methods: Qualitative data were collected in diverse ways, mainly from interviews. Seventeen Chinese preservice physical education teachers were selected as participants by a purposive sampling technique. The interview questions focused on participants’ motivation, expectations, and experiences in social media usage. Grounded theory was used to analyze the data by ROST CM and Nvivo 12. (3) Results: The perception of social media among teachers includes three subsidiary categories made up of 10 sub-categories, 70 concepts, and 307 labels. The three categories are (a) value perception, including the perspective of intelligent function, interaction, and rich information, (b) risk perception, involving psychological risk, information risk and privacy risk and (c) overall perception, like development trends, current status and basic elements. (4) Conclusions: Chinese preservice physical education teachers perceive social media as having similarities and differences compared to other countries. Future research should consider a large sample survey to revise and verify the initial exploration of perception and study diverse groups of teachers’ perceptions of social media.

## 1. Introduction

Social media has emerged as a prevalent and widely used form of new media, embraced by individuals worldwide, with a special focus on the youth demographic. As for the youngest cohort of physical education teachers, preservice physical education teachers, who possess an insatiable curiosity but lack teaching guidance and experience, social media presents a double-edged sword of opportunities and challenges. On the one hand, social media offers various benefits, such as free access to professional knowledge, skills, and opinions, serving as a primary source for news retrieval, and providing avenues for global communication. As a result, social media has become an essential tool for professional development among physical education teachers [1]. On the other hand, the growing prevalence of fake news on social media is a concern, especially in fields such as physical education and health [2]. The dissemination of false information through social media can harm the reputation and credibility of preservice physical education teachers and have a negative impact on their professional standing. Therefore, it is crucial for preservice physical education teachers to be mindful of both the potential consequences of fake news and the benefits of utilizing social media in a responsible and informed manner.

The perception of social media among physical education teachers has received increasing attention in recent years, given its potential impact on their professional development. In order to gain a deeper understanding of social media in physical education, several studies have employed a grounded theory approach. This method is growing in popularity, especially in physical education. A few examples of studies that used grounded theory include Pummell, Harwood, and Lavallee (2008), who explored perceptions of within-career transitions among equestrian event riders [3], Goodyear et al. (2019), who studied the characteristics of a Twitter-based professional learning community called #pechat among physical education teachers [4], McNamara et al. (2021), who found that physical education teachers were more likely to engage in communication through social media for learning activities [5], and Hyndman and Harvey (2019, 2020), who examined preservice teachers’ perceptions of using Twitter in health and physical education teacher education [6,7]. The study by Goodyear et al. (2019) provides valuable insights into the potential of social media to support the professional development of physical education teachers, while McNamara et al. (2021) found that physical education teachers were more likely to engage in communication through learning activities on social media. Hyndman and Harvey (2019, 2020) determined that preservice teachers saw many values in using Twitter for health and physical education teacher education, such as autonomy, relatedness, and competence, but also raised concerns about undesired Twitter users and navigating the platform. As we can observe, the understanding of the perception of social media among physical education teachers is becoming increasingly comprehensive, however, the lack of representation from eastern countries makes the goal difficult to achieve.

Previous research has made a huge contribution to the field of social media and physical education teachers’ professional development, while the current investigation includes limitations related to the sample that should be acknowledged in interpreting the results [8]. The geographical location of previous research is unclear. Moreover, most of the research conducted and collected data is from the United States [1,5,7,8,9,10,11,12,13,14,15,16], followed by Australia [6,17], Spain [18,19,20], Ireland [21,22], and the United Kingdom [23,24,25,26], while only three related research included developing countries [27,28,29]. The research from eastern countries sees social media only as a data collection platform instead of in the context of teachers’ professional development [27] or has used questionnaire surveys to conduct the research [28]. A questionnaire survey is a statistical study to test a theory on social phenomena or human problems. It measures variables with numerical data and analyzes results to validate a theory [30,31]. To some extent, the questionnaire suits the research topic of teachers’ perceptions. Unfortunately, the perceptions among teachers are so complex and interwoven that they cannot be reduced to isolated variables and presented in detail. The perception of social media is complex, making it necessary to use a theoretical model to better understand it. Among various methods available, grounded theory is considered the best choice. Grounded theory allows researchers to create a theory based on participants’ experiences, perspectives, and behaviors, providing insight into how people perceive and make sense of social media. With its inductive data analysis, grounded theory enables researchers to develop a theory specific to the participants’ experiences and allows for an iterative process of data collection and analysis, leading to a deeper understanding of participants’ perceptions [32]. There is limited knowledge regarding the perception of preservice physical education teachers towards social media in China. If teachers lack a thorough understanding of social media, they may only use it for leisure or personal purposes rather than for their professional development. To address this, exploring a theoretical model of preservice physical education teachers’ social media perceptions could inform their understanding of social media and provide a scientific framework for effective professional development. Also, the potential for generalizability from an in-depth study was addressed. A wider sample could have provided further insights [4] and explored similarities and differences in different groups of teachers [9]. China is one of the biggest countries training preservice physical education teachers. Specifically, in mainland China, there are 1423 colleges and 1265 universities totaling 2688 higher education institutions, and 317 of these universities offer physical education teacher education (PETE) programs for preservice physical education teachers [33].

Chinese preservice physical education teachers are a large group of members of social media users for their professional development and use different social media platforms compared to teachers from western countries. However, little is known about Chinese preservice physical education teachers’ perceptions of their own social media usage. It is obvious that different culture differences exist among different countries, which suggests a need to explore the differences instead of totally copying the experience from other countries. Such information is theoretically significant given the paucity of knowledge about this professional group. It can also have practical implications for future planning for international collaborative professional development and identifying the specific challenges or professional needs among preservice physical education teachers. Given the scholarly attention afforded to social media to promote professional development among physical education teachers, it is necessary to fill the research gap, especially in terms of evidence, population, and methodology. To ameliorate these weaknesses, we used grounded theory, which is well-suited for perception research because it allows researchers to develop a theory that is grounded in the experiences and perspectives of the participants, provides an inductive approach to data analysis, and allows for an iterative process of data collection and analysis. We also focus on a Chinese sample, which has the largest population of physical education teachers.

In summary, the utilization of social media among preservice physical education teachers has increased, yet the comprehension of their perceptions regarding the usage of social media remains limited, particularly in China. The aim of this study is to address this gap by exploring preservice physical education teachers’ perceptions of social media and developing a theoretical model to capture these perceptions. This study seeks to contribute to the advancement of knowledge concerning social media in physical education teacher education, providing valuable insights for teacher educators and educational institutions.

## 2. Materials and Methods

As mentioned above, pre-service physical education teachers frequently use social media for various purposes. Nevertheless, their perceptions of social media are not well-understood, and this lack of understanding could impact their ability to use social media appropriately in their future professional endeavors. Thus, our research question is what is the perception of social media among Chinese preservice physical education teachers? Our research objective is to develop a theoretical model of social media perception among Chinese preservice physical education teachers. Specifically, grounded theory is an effective approach for developing the perception model of social media, as it is based on participants’ experiences, provides inductive data analysis, and allows for the refinement of understanding as data is collected. The basic logic of grounded theory is to condense data from the bottom up and establish a theory [34]. Researchers do not make assumptions beforehand but instead initiate data induction analysis directly. As a result, the relevance and validity of the data must be ensured during data collection.

This study collected data through online posts, literature, and interviews. First, a literature search was conducted using domestic and international databases such as CNKI (China National Knowledge Infrastructure) and Web of Science to gather information on the perception model of social media among pre-service physical education teachers. A total of 48 Chinese and 203 English relevant articles were obtained and analyzed to extract preliminary concepts and support the theoretical framework developed in the text. Additionally, online posts related to physical education, such as #pechat and #pe, were collected from popular social media platforms (WeChat, Twitter, TikTok, etc.). Finally, semi-structured interviews were conducted with 17 pre-service physical education teachers. The details of the interviews are described below.

### 2.1. Participants

It is important to note that the appropriate sample size in grounded theory research is determined by the data and not by a specific number of participants. The sample size should be sufficient to generate a rich and diverse data set that allows the researcher to develop a theory that is grounded in the data. The sample size was based on the criterion of theoretical saturation; that is, no new relevant data emerged regarding a category, categories were well-developed, and relationships among categories were established and validated [32]. In the field of physical education, studies demonstrate that a sample size of 10–20 participants can be sufficient in grounded theory research [35], particularly when the aim is to generate a rich and detailed understanding of a particular phenomenon. Hence, 17 participants were selected to be interviewed, which meets the requirements of grounded theory.

Following ethical approval, participants were recruited through a combination of purposeful and snowball sampling [36]. Participants were sourced from every class level and every type of university, and every part of China (East–North, East–South, West–North, West–South). Hence, it can be inferred that the participants have a plausible representation of the target population. The study employed three revised inclusion criteria from Goodyear (2019) [4] to select participants: (1) current active use of social media (defined as frequent daily use); (2) a minimum of five years of experience using social media; and (3) utilization of social media for multiple purposes, including professional development. In order to preserve the anonymity of participants, each was assigned a numerical identifier from 1 to 17 (refer to Table 1 for details).

### 2.2. Interview

Ethical approval was approved by the research ethics committee of the authors’ institute. A conversational, in-depth interview was guided by six interview questions (see Table 2). These interview questions were modified from a previous study [4,11,34]. The interview questions were initially developed based on the goals of the study and were discussed with three professors in the field of physical education teacher education (PETE). After several revisions, the final interview outline included six open-ended questions. These questions covered two main topics: (1) participants’ perceptions of the characteristics and uses of social media and (2) their perceptions of the challenges, benefits, and preferences related to using social media. Due to the COVID-19 pandemic, the 17 individual semi-structured interviews were conducted online. At the beginning of each interview, we followed previous studies and established a trusting relationship with the participants. This was done in order to ensure that the participants felt comfortable and willing to share their thoughts and experiences with us. By creating a positive and open environment, we were able to gather rich and valuable data that helped us better understand the topic at hand [37]. To put the participants at ease, we engaged them in casual conversation. This included discussing hot topics on microblogs, explaining funny emojis on social media, and highlighting the importance of social media usage. The interviews lasted an average of 60 min, and all participants agreed to have their interviews recorded. This allowed us to capture their responses accurately and in full, which was crucial for our analysis.

### 2.3. Data Analysis

When all interviews were completed, the recordings were transcribed verbatim by ROST CM 6 software [38], and NVivo 12 [39] was used to store and manage the data. The data were analyzed using constant comparative analysis, and the transcripts of each interview were carefully reviewed. Initial evaluation of the data was followed using grounded theory, including the steps of open coding, axial coding, and selective coding to analyze the qualitative data. This systematic approach allowed us to identify key themes and patterns in the data, which provided insights into participants’ perceptions of social media and its use in the context of physical education teacher education [40]. The data for this paper were collected from a variety of sources, including literature on social media perception among physical education teachers, social media posts related to physical education, and semi-structured interviews with preservice physical education teachers. These data were then analyzed using a coding process to identify key themes and patterns, which allowed us to form a comprehensive understanding of participants’ perceptions of social media in their professional fields. The resulting research provides insights into the potential challenges and benefits of using social media in Chinese physical education teacher education, as well as suggestions for future research on physical education teacher education [41].

#### 2.3.1. Open Coding

Open coding refers to the initial conceptualization, further conceptualization and categorization of raw empirical data [40] related to preservice physical education teachers’ social media perceptions. The open coding process was used to identify key concepts and categories related to preservice physical education teachers’ perceptions of social media. This involved capturing keywords that appeared in primary sources related to these perceptions and using native words from empirical sources to form free nodes for preservice physical education teachers’ social media perceptions. This allowed us to build a comprehensive understanding of the key themes and patterns in the data and to develop a theoretical framework for our research.

Step 1: Labeling and Conceptualization. Through the open coding process, we identified 307 labels related to preservice physical education teachers’ perceptions of social media. These labels were then subjected to constant comparative analysis, in which they were compared with each other to identify key themes and patterns. This process, guided by grounded theory, allowed us to condense and refine the labels, resulting in 70 exclusive concepts that captured the essence of preservice physical education teachers’ social media perceptions. These concepts provided a solid foundation for our theoretical framework and helped us to better understand the data we had collected before (see Table 3 and Table 4).

Step 2: Categorization. Using the same process of constant comparative analysis, we further classified and compared the 70 concepts we had identified. This allowed us to identify 10 subsidiary categories that captured the key themes and patterns in the data. These categories included Media Interface Perception, Privacy Risk Perception, Information Value Perception, Basic Element Perception, Current Status Perception, Information Risk Perception, Intelligent Function Perception, Development Trend Perception, Interactive Value Perception, and Psychological Risk Perception. These categories provided a more detailed and nuanced understanding of preservice physical education teachers’ perceptions of social media and helped us to develop a comprehensive theoretical framework for our research. (Seen in Table 5).

#### 2.3.2. Axial Coding

Axial coding refers to the categorization and comparison of various concepts or categories related to preservice physical education teachers’ perceptions of social media. This process involves breaking down and reorganizing the free nodes of these perceptions to extract the main concepts or categories. In axial coding, we dismantled the free nodes of preservice physical education teachers’ social media perceptions and reorganized them in a way that allowed us to analyze and conceptualize these perceptions in a more systematic and comprehensive manner. This helped us to identify key themes and patterns in the data and to develop a more robust and nuanced theoretical framework for our research.

Through further constant comparative analysis, we condensed the 10 subsidiary categories identified in the axial coding process into three main subcategories: Overall Perception, Value Perception, and Risk Perception. Each of these subcategories included different subsidiary categories from the original set. Specifically, the “Overall Perception” subcategory included “Interface Perception”, “Element Perception”, “Current Status Perception”, and “Development Trend Perception”; the “Value Perception” subcategory included “Information Value Perception”, “Interactive Value Perception”, and “Intelligent Function Perception”, and the “Risk Perception” subcategory included “Psychological Risk Perception”, “Privacy Risk Perception”, and “Information Risk Perception”. These subcategories provided a more refined and focused view of preservice physical education teachers’ perceptions of social media and helped us to develop a more comprehensive and nuanced theoretical framework for our research. (See Table 6 for details).

#### 2.3.3. Selective Coding

Selective coding is the process of analyzing the links between the different main concepts or categories [40] of pre-service PE teachers’ social media perceptions and making continuous comparisons to further uncover the ‘core categories’ that can unify all pre-service physical education teachers’ social media perception concepts or categories. Selective coding is the final stage of the grounded theory analysis process, in which the data are analyzed in relation to the key themes and patterns identified in the earlier stages of coding. This process involves making connections between the data and the concepts and categories that have been identified and using memos and other analytical tools to develop a comprehensive theoretical framework for the research. In the case of this study, selective coding helped us to understand the key factors that influence preservice physical education teachers’ perceptions of social media and to develop a theoretical model that captures the complex and nuanced nature of these perceptions.

Through the continued examination of the 10 subsidiary categories and three subcategories and the constant comparison of the original data, we were able to identify a core category for our research: “preservice physical education teachers’ social media perceptions”. This core category served to integrate all of the other categories, concepts, and labels that we had identified through the coding process, providing a comprehensive and holistic view of the research topic. The core category allowed us to develop a theoretical framework that captured the key themes and patterns in the data and provided a solid foundation for our analysis and interpretation of the results.

In summary, the data collected for this study were rich and diverse, including interviews, literature, and social media posts. These data were labeled, conceptualized, and categorized through the open coding process. This allowed us to identify key themes and patterns in the data and to group the categories into subcategories through axial coding. Finally, through selective coding, we were able to develop a comprehensive theoretical framework for our research, consisting of a core category (“preservice physical education teachers’ social media perceptions”), three subcategories (Overall Perception, Value Perception, and Risk Perception), 10 subsidiary categories, 70 concepts, and 307 labels. This theoretical framework provided a solid foundation for our analysis and interpretation of the data and allowed us to better understand the complex and nuanced nature of preservice physical education teachers’ perceptions of social media.

In conclusion, the data collected for this study were initially conceptualized, then deeply conceptualized and categorized, and finally integrated into a theoretical framework through the grounded theory analysis process. This allowed us to identify key themes and patterns in the data and to develop a comprehensive understanding of preservice physical education teachers’ perceptions of social media. The resulting theoretical model provides a solid foundation for future research on this topic and offers insights into the potential challenges and benefits of using social media in physical education teacher education.

### 2.4. Trustworthiness

Guided by grounded theory, theoretical saturation is related to trustworthiness [40]. To ensure the reliability and validity of our coding process, we performed several tests and checks. This included conducting a coding consistency test to reduce subjectivity and increase confidence in the coding results. For this test, we selected five profiles at random from the original data and asked two researcher assistants who were familiar with the use of Nvivo 12 software to simultaneously code the sample back-to-back. We then calculated the coding repetition rate for the sample and compared it to the results obtained from the initial coding. Additionally, we conducted a theoretical saturation test using the remaining five interview profiles as reference materials. This allowed us to explore whether new categories and relationships could be identified from these profiles and whether the theoretical model constructed for the study had reached theoretical saturation. Overall, these tests and checks ensured the reliability and validity of our coding process and the resulting theoretical framework.

## 3. Results

Through our analysis of the data, we identified three major themes that capture preservice physical education teachers’ perceptions of social media: (1) Value Perception, (2) Risk Perception, and (3) Overall Perception. Each of these themes represents a different aspect of preservice physical education teachers’ perceptions, and together they provide a comprehensive understanding of the complexities and nuances of these perceptions. Under each of these main themes, we identified numerous subthemes that further elaborated on the key concepts and categories identified in the data. These subthemes provided a more detailed and refined view of preservice physical education teachers’ perceptions of social media and helped us to better understand the factors that influence these perceptions (Figure 1).

### 3.1. Value Perception: Multiple Motivations for Social Media Usage

The theme of “Value Perception” refers to the reasons why preservice physical education teachers use social media and the expectations they have for its usage. This theme included subsidiary categories such as “Information Value Perception”, “Interactive Value Perception”, and “Intelligent Function Perception”. These categories captured the key factors that influence preservice physical education teachers’ decisions to use social media, as well as the benefits and opportunities that they expect to gain from its usage. Overall, the theme of Value Perception provided insight into the motivations and expectations of preservice physical education teachers in relation to social media.

#### 3.1.1. Information Value Perception

One of the most frequent subthemes under the theme of Value Perception was “Information Value Perception”. This subtheme referred to the value that preservice physical education teachers place on the wealth of information and resources available on social media. For example, many participants mentioned that social media provides them with access to a wide range of educational materials and resources that they can use to enhance their teaching practices. They also mentioned that social media allows them to connect with other teachers and experts in the field, which provide valuable insights and support for their professional development. Overall, the subtheme of Information Value Perception highlighted the importance of social media as a source of information and knowledge for preservice physical education teachers:


*The professional knowledge we can learn in school is relatively comprehensive and systematic. However, to some extent, it is not always what we want to learn, hence things we learnt in school are conventional and not very practical.*
(Participant 2)

One of the benefits of using social media for preservice physical education teachers is that it allows them to access cutting-edge knowledge and specialized information that is not readily available in traditional classroom settings. Through their interactions on social media, preservice physical education teachers may become aware of their own limitations and gaps in knowledge, which can motivate them to learn and grow as professionals. For example, some participants mentioned that they were able to learn about new teaching strategies, research findings, and trends in the field of physical education through social media, which helped them to improve their teaching practices and stay up-to-date with the latest developments in their field. Overall, the subtheme of Information Value Perception highlighted the potential of social media as a tool for professional growth and development among preservice physical education teachers.


*If we cannot update our knowledge and physical education philosophy in a timely manner, we are likely to fail to engage students in physical education, resulting in students increasingly finding physical education classes boring, as they can search for such knowledge content without the need for a teacher, and it is likely that the knowledge taught is what students have learned long ago.*
(Participant 3)

Overall, the theme of Value Perception revealed that preservice physical education teachers place a high value on the wealth of information and resources available on social media. This is particularly true for those who are motivated to learn and grow as professionals and who see social media as a valuable tool for accessing knowledge and expertise in their field. In this respect, the availability of rich and diverse information on social media emerged as the most significant consideration for preservice physical education teachers in relation to their use of social media.

#### 3.1.2. Interactive Value Perception

Participants also discussed the interaction as an enabler for social media usage. Regarding interactive perception, social media utilizing social networking technologies provide new opportunities for initiating “webs of enhanced practice” [42], where teachers around the globe can engage in collegial collaborations that enhance motivation for professional development.

Another key subtheme under the theme of Value Perception was “Interactive Value Perception”. This subtheme referred to the value that preservice physical education teachers place on the social and interactive aspects of social media. For example, many participants mentioned that social media provides them with access to online communities of fellow teachers and educators, which can serve as a valuable source of support, collaboration, and networking. These online communities can provide a sense of connection and belonging that may not be available in the offline world and can help preservice physical education teachers to feel less isolated and more engaged in their profession. Additionally, some participants mentioned that social media provides them with opportunities to connect with peers and experts from different parts of the world, which can broaden their perspectives and enrich their learning experiences. Overall, the subtheme of Interactive Value Perception highlighted the importance of social media as a platform for building community and fostering collaboration among preservice physical education teachers.


*I do feel that social media is very widely used nowadays, but in fact there is a big difference in the use of each person’s application. For example, everyone in our dormitory knows about Shake Voice, but several other students spend most of their time searching for some meaningless funny videos and amusing each other with the so-called stems, while I think that as a student most of my time should be spent on studying. So I spend most of my time on social media sharing videos I’ve learnt about jumping rope, and I’ve also joined a book sharing group to warn myself not to be complicit and to spend more time studying outside of class.*
(Participant 1)

#### 3.1.3. Intelligent Function Perception

Participants also highlighted Intelligent Function Perception. The subtheme of Intelligent Function illustrates the relationship between technology and social media. Digital technology is regarded as an extension of self, and social media is a primary mode of communication and social engagement [43].

Another subtheme under the theme of Value Perception was “Intelligent Function Perception”. This subtheme referred to the value that preservice physical education teachers place on the advanced technology and features that are integrated into social media platforms. For example, many participants mentioned that social media platforms include a range of intelligent functions, such as language translation, automatic sourcing of posts, and information storage and management, which can enhance their social media experience and make it more efficient and effective. Additionally, participants mentioned that the use of mobile devices and other technologies has made it easier and more convenient to access social media and that the well-rounded functions of these platforms fully meet their needs for professional development and communication. Overall, the subtheme of Intelligent Function Perception highlighted the importance of technology and advanced features in shaping preservice physical education teachers’ perceptions of social media.

### 3.2. Risk Perception: Concerns for Its Safety and Effectiveness

Despite reporting the perception of values that preservice physical education teachers perceive in social media, they also reported concerns about the risks and challenges associated with its use. The theme of “Risk Perception” captured the potential hazards and drawbacks that preservice physical education teachers perceive in their use of social media. This theme included three subsidiary categories: “Psychological Risk Perception”, “Information Risk Perception”, and “Privacy Risk Perception”. These categories captured the key factors that influence preservice physical education teachers’ perceptions of the risks associated with social media, as well as the potential consequences of these risks. Overall, the theme of Risk Perception provided insight into the challenges and concerns that preservice physical education teachers face in relation to their use of social media.

#### 3.2.1. Psychological Risk Perception

Participants expressed concern about the psychological risks associated with perception on social media. They noted that online posts on social media can lead to a phenomenon known as “context collapse,” in which a small mistake can cause significant changes to an individual’s identity in real life. This can be especially damaging for those who rely heavily on their online presence for personal and professional interactions:


*To maintain a professional image as a teacher in social media, we always need to be careful about what we say and do. Especially sometimes it might have a negative impact on the reality life (e.g., unemployed, accusation, damage the image of teachers or local school). As a result, I tend to reduce unnecessary social media interactions for avoiding misunderstandings.*
(Participant 6)

#### 3.2.2. Information Risk Perception

The final subtheme that emerged from the data was that preservice physical education teachers shared concerns about the limitations of information resources, which they referred to as “Information Risk Perception.” In this subtheme, participants frequently discussed concerns about the accuracy and professionalism of social media posts. They expressed a lack of trust in the information found on social media and the potential consequences of relying on such sources for professional development and decision-making. Some participants suggested that professional organizations and institutions should provide more reliable and systematic sources of information to help address this issue.


*Sometimes I feel overwhelmed about the posts related to physical education or sports, I can’t identify if it is true or not. For example, I found conflicting explanations of physical skill training methods in different sports social media platforms. So what do you expect this information? Trust or not? It is hard to say.*
(Participant 10)


*People send sports-related posts in social media are always not very serious, the same time, I find it difficult to screen the authenticity of basic sports theory or specific sports data in social media. So, I prefer to ask my teachers at university, it’s time consuming to find sports-related resources that fits my specific questions in social media.*
(Participant 12)

#### 3.2.3. Privacy Risk Perception

As the popularity of social media platforms such as WeChat and Micro-blog continues to grow, our privacy has become increasingly vulnerable to surveillance and even commodification. As preservice physical education teachers upload personal information to social media, more and more users are able to access this rich data at will. However, there is a lack of transparency and control over how this data is used and by whom, which can create a sense of anxiety and distress for preservice physical education teachers who use social media for professional development. This concern highlights the need for greater awareness and caution when using social media for professional purposes:


*It is true that there are a lot of people sharing physical education knowledge and skills on social media, and although we are also physical education students and are still relatively more knowledgeable about physical education than other users, because we are students at X school, we are expected to represent the school in our words and actions, and as a teacher educator, we are also expected to take an exemplary role and should not make random comments.*
(Participant 7)

### 3.3. Overall Perception: Uniqueness of Social Media for Teachers

Overall Perception refers to the general understanding of preservice physical education teachers’ perception of social media. This overarching view shapes their decisions on whether to use social media and for what purposes. Participants shared their imaginations of what social media looks like, and these imaginations were clustered into four subthemes: (i) media interface perception, (ii) basic element perception, (iii) current status perception, and (iv) development trend perception. These subthemes provide insight into preservice physical education teachers’ attitudes and beliefs about social media and its role in their professional development.

#### 3.3.1. Media Interface Perception

The definition of “media interface perception” is how users obtain and process information through the interface of social media platforms. This was also the initial impression that preservice physical education teachers had of social media before deciding to use it for professional development. In terms of media interface perception, the prevalent discussion among participants related to perceiving the quality of images and length of text on social media:


*For our physical education teachers, video is essential. Learn sports skills always related movement, so I prefer to search videos more than e-books. In social media, graphic videos related to sports or teaching practices always have high definition picture quality, clear and easy to understand.*
(Participant 11)


*We physical education teaching should be creative, so sometimes I like have unique design for my own social media page. Social media fits my need, I think the vivid and colorful interface matches the aesthetics of all pre-service physical education teachers.*
(Participant 14)

#### 3.3.2. Basic Element Perception

The subtheme of “basic element perception” highlights the concerns that preservice physical education teachers have about using social media for professional development. This subtheme includes their concerns about the financial cost and time investment required to access high-quality knowledge in physical education, as well as the information available on social media:


*As for the curriculum in universities, we are hardly find extra time to learn knowledge and skills comprehensively. Social media is such a time-saving way for us because I find it very easy to operate the video clips and other applications needed to share motor skills.*
(Participant 15)


*Buying a textbook or specialised books on sports disciplines are always expensive for our students, while I feel that paid sports resources in social media are affordable. But sometime the resources might be others posts by taking picture or recorded video with limited shooting angles. Such posts might affect the learning effect of technical movements.*
(Participant 9)

#### 3.3.3. Current Status Perception

The subtheme of “current status perception” reveals the participants’ views on the popularity of social media in university and outside of the university, both online and offline. This subtheme suggests that preservice physical education teachers recognize the widespread use of social media in their personal and professional lives and may be considering how to integrate these platforms into their professional development:


*There is no doubt that social media is a platform that welcome to everyone. We physical education teachers are such a group. While I still found physical education majors and teachers are using social media all the world!*
(Participant 2)


*Since I find that there are many users and even many authoritative physical education experts in social media. Those people are all interested in exercise and fitness, I think social media has solved exactly the paradox of learning and training that many pre-service physical education teachers faced.*
(Participant 5)

#### 3.3.4. Development Trend Perception

Participants also reflected on the future of social media and its potential impact on the field of physical education. Some participants mentioned that it may take time for new technologies to be fully integrated into regular physical education teaching in schools. Additionally, there were many participants who expressed ambivalent views about the use of social media in physical education, suggesting that there is still some uncertainty about the role of these platforms in the field:


*I feel (pause for almost a minute) it’s hard to say, for myself of course I think it’s good that social media is used in the classroom to save a lot of time in lesson preparation and students are exposed to a very rich knowledge of physical education, not just learning motor skills, but especially to make students aware of the value of physical education. But because physical education is given limited attention in schools, if schools were to make IT universal it would require a huge investment and could also cause opposition from parents. If that day does come then schools will most likely give the opportunity to subjects such as languages, maths, and foreign languages first, and it will take time for physical education lessons (social media applications) to become widespread.*
(Participant 11)

## 4. Discussion

According to the data collected from 17 preservice physical education teachers from various locations and types of universities in China, the present study investigated preservice physical education teachers’ perception of social media usage by uniquely employing Nvivo 12 software and grounded theory. Our findings answered three questions: (a) What aspects of social media do preservice physical education teachers find valuable? (b) What do preservice physical education teachers perceive the characters and elements of social media? (c) What kind of challenges have preservice physical education teachers faced in social media usage towards their professional development? By answering these questions, these reflections highlight the need for further research and discussion about the potential benefits and drawbacks of using social media for professional development in physical education.

### 4.1. Complex and Multi-Faceted Relationships among all Three Categories of Perception

The interplay between Overall Perception, Value Perception, and Risk Perception in social media is a complex and multi-layered phenomenon, as demonstrated below.

The relationship between Overall Perception and Value Perception in social media can be significant. Overall Perception, as the general understanding of preservice physical education teachers’ view of social media, can influence their Value Perception of social media. For instance, if preservice physical education teachers have a positive Overall Perception of social media, they may view it as valuable in terms of helping them stay connected with peers in their physical education teacher education program, and they may view the benefits of social media as increased social connectedness and reduced feelings of isolation. In contrast, if they have an overall negative perception of social media, they may view it as having limited value and see the risks as outweighing the benefits. Similarly, the relationship between Overall Perception and Risk Perception in social media is also important. Participants’ Overall Perception of social media can impact their views on the risks associated with its use. If preservice physical education teachers have a positive Overall Perception of social media, they may be less likely to view the risks as significant and may be more willing to continue using social media. Conversely, if they have an overall negative perception of social media, they may view the risks as more substantial and may be less likely to use it. In conclusion, the Overall Perception of preservice physical education teachers plays a critical role in shaping their Value Perception and Risk Perception of social media. A positive Overall Perception can lead to positive value and Risk Perceptions, while an overall negative perception can lead to a negative value and Risk Perceptions.

The Value Perception of social media can also impact views on its associated risks. If preservice physical education teachers believe that social media has a positive impact on their future careers, they may be inclined to view the risks as insignificant or manageable. Conversely, if they view social media as having limited value or if the risks outweigh the benefits, they may perceive the risks of social media use as more substantial. Similarly, Risk Perception of social media can affect how preservice physical education teachers engage with it. If someone believes that social media poses a significant danger to their privacy or well-being, they may reduce their usage or be more cautious when interacting with others in the field of physical education. For example, physical education teachers can post videos of exercises and activities, while health organizations can share infographics and illustrations to educate people on healthy habits. On the other hand, if they view the risks as manageable, they may continue to use social media without much thought to the potential risks. In the YouTube context, if preservice physical education teachers view instructional videos on physical education and health topics shared on YouTube as valuable, they may view the risks associated with using the platform, such as exposure to inappropriate content, as manageable and continue to use it for learning and professional development. Conversely, if they view the information shared on YouTube as less valuable, they may be less likely to use the platform or may view the risks as greater, potentially limiting their use of the platform.

In summary, the relationship between Overall Perception, Value Perception, and Risk Perception in social media is intricate and multi-faceted and can greatly impact how preservice physical education teachers use and engage with it. Preservice physical education teachers can learn that the way they perceive and engage with social media is influenced by their Overall Perception of it, as well as their perception of its value and the associated risks. They can use this information to carefully consider their use of social media, taking into account how it may impact their professional development and approach to teaching physical education. By being aware of the intricate and multi-faceted relationship between these factors, preservice physical education teachers can make informed decisions about using social media to advance their knowledge and skills and stay updated on the latest trends and developments in the field.

### 4.2. Social Media Reflects Preservice Physical Education Teachers’ Needs in a New Era

Our findings suggest that preservice physical education teachers perceive social media meets their professional needs in a new era.

Based on the theme of “Value Perception” in the discussions of preservice physical education teachers, it can be inferred that these teachers have a professional development need for gaining new information, opportunities for interactive communication and learning, and access to intelligent functions through the use of social media. These categories reflect the factors that drive their decision to use social media and the benefits they hope to gain from its usage. Thus, it can be concluded that preservice physical education teachers have a professional development need for exploring the potential of social media in advancing their knowledge, skills and overall professional growth. Based on the theme of “Value Perception” in the discussions of preservice physical education teachers, it can be inferred that these teachers have a professional development need for gaining new information, opportunities for interactive communication and learning, and access to intelligent functions through social media. These categories reflect the factors that drive their decision to use social media and the benefits they hope to gain from its usage. Thus, it can be concluded that preservice physical education teachers have a professional development need for exploring the potential of social media in advancing their knowledge, skills and overall professional growth.

Based on the Risk Perception we explored in this study, it can be inferred that the professional development needs among preservice physical education teachers include understanding and managing the potential risks associated with social media usage. This includes managing psychological risks, understanding the risks related to information, and managing privacy risks. One of the primary concerns identified is the understanding and management of potential risks associated with social media usage. This encompasses a wide range of potential risks, including psychological risks, information risks, and privacy risks. Regarding psychological risks, preservice physical education teachers need to be aware of the impact that social media can have on their well-being, such as the potential for cyberbullying, exposure to negativity, and the impact of social comparison. These teachers must also be equipped with strategies to manage these risks, such as setting healthy boundaries, seeking support, and engaging in self-care practices. In terms of information risks, preservice physical education teachers must understand the potential for misinformation, propaganda, and other forms of false information to be spread via social media. They need to be equipped with the skills to identify and verify the information and understand the importance of source evaluation. Finally, privacy risks are also a major concern for preservice physical education teachers. They must understand the potential for personal information to be shared or leaked on social media, as well as the dangers of oversharing personal information. They need to be equipped with strategies for managing their privacy, such as understanding privacy settings and being mindful of the information they share.

Based on the Overall Perception explored in our study, it can be inferred that the professional needs among preservice physical education teachers include a comprehensive understanding of social media and its role in their professional development. The Overall Perception of social media among preservice physical education teachers shapes their decisions on whether to use it and for what purposes. The study revealed that preservice physical education teachers have concerns about the financial cost and time investment required to access high-quality knowledge in physical education through social media, as well as the quality of information available on these platforms. The sub-themes of media interface perception and basic element perception provide insight into preservice physical education teachers’ attitudes and beliefs about social media. The media interface perception refers to how users obtain and process information through the interface of social media platforms and the initial impression that preservice physical education teachers have of social media. The basic element of perception highlights the concerns that preservice physical education teachers have about using social media for professional development, including the financial cost and time investment required to access high-quality knowledge. Furthermore, the study revealed that participants had mixed views about the future of social media and its potential impact on the field of physical education. Some participants expressed uncertainty about the role of social media in physical education, suggesting that there is still some ambiguity about its use in the field.

### 4.3. Teachers’ Perception of Social Media Followed Philosophical Foundations

“Goffman’s theory of self-presentation”, “Bourdieu’s theory of social capital”, “Sartre’s existentialism”, and “Heidegger’s theory of shared Theories” fit the connotations of the perception model we generated. These theories provided ideas for the process of generating this perception model of social media among preservice physical education teachers:

Self-representation theory is the ‘action’ basis for the generation of pre-service physical education teachers’ social media perception. Self-representation theory can be applied to the three categories of perception (Overall Perception, Value Perception, and Risk Perception) in social media usage among preservice physical education teachers. The theory suggests that individuals use self-representation to create a self-image that they present to others in order to gain social validation. For example, applying this to the Value Perception, preservice physical education teachers may use social media to present a positive self-image as knowledgeable, up-to-date, and connected professionals in the field of physical education. They may also use it to gain information and new ideas for their professional development or to interact with other professionals in their field to build relationships and networks. In applying this to Risk Perception, preservice physical education teachers may be concerned about the potential negative impact that social media usage can have on their self-image. They may worry about psychological risks, such as exposure to negativity or bullying, information risks, such as the spread of misinformation, or privacy risks, such as the potential exposure of sensitive information. These concerns may lead them to limit their social media usage or be more cautious in how they present themselves online. Finally, applying this to the Overall Perception, preservice physical education teachers may see social media as a tool that can serve both their self-representation and professional development needs but also understand the potential risks associated with its use. They may weigh the benefits and drawbacks of social media usage in light of their personal goals and values to make informed decisions about their engagement with the platform.

Social capital theory is the ‘zeitgeist’ basis for the generation of preservice physical education teachers’ perception of social media. The “Value Perception” theme highlights the motivations and expectations of preservice physical education teachers for using social media for their promising professional development. According to social capital theory, social media can provide opportunities for individuals to gain valuable connections and information that can increase their overall social capital. In this case, the preservice physical education teachers use social media to gain access to information that is relevant to their profession, such as pedagogical content, health promotion and body image, which might enhance their future teaching practice. In contrast, the “Risk Perception” theme highlights the potential hazards and drawbacks that preservice physical education teachers perceive in their use of social media. According to social capital theory, the risks associated with social media use can reduce an individual’s overall social capital. In this case, the preservice physical education teachers may be concerned about the potential consequences of using social media, such as psychological harm in their relationship with classmates in the physical education teacher education program, privacy violations, and information security, which can negatively impact their professional development via social media. The “Overall Perception” theme encompasses both value and risk perceptions and provides insight into the complex and multi-faceted relationship between these two factors. According to social capital theory, the Overall Perception of social media use can greatly impact an individual’s ability to gain social capital. In this case, the preservice physical education teachers’ Overall Perception of social media will influence their decisions to use it and their expectations for its usage, which can shape their professional development outcomes.

Shared world theory is the ‘driving force’ behind the formation of social media perceptual structures for pre-service physical education teachers. According to Heidegger, our existence in the world is characterized by care and concern for others [44], further supporting the conclusion that social media helps to reduce teachers’ sense of isolation and increase a sense of belonging [45,46,47]. Heidegger’s theory of shared theories suggests that individuals construct their own understanding of reality through their interactions and experiences with others. In the context of the three perceptions among preservice physical education teachers (Value Perception, Risk Perception, and Overall Perception), this theory suggests that their perceptions of social media are shaped by their interactions and experiences with others who use it. For example, if a preservice physical education teacher has positive experiences and interactions with other teachers who use social media, they are likely to have a positive Overall Perception of it and see its value for their professional development. On the other hand, if they have negative experiences and interactions, they may perceive more risks associated with using social media and be less likely to see its value. Therefore, the theory of shared theories highlights the role of social interactions in shaping an individual’s perceptions of social media.

### 4.4. Limitations and Future Research

Overall, this study suggests that while social media has the potential to be a valuable tool for professional development in physical education, it is also influenced by a range of objective factors. Given the limited sample size of this study, it is recommended that future research incorporate a larger sample to validate and expand upon these findings. Moreover, future studies should explore the theoretical model of social media perception among diverse groups, including in-service physical education teachers in primary and secondary schools, as well as university-based physical education teacher educators. Such research would yield a more comprehensive understanding of the role of social media in physical education and inform the development of effective strategies for using these platforms in professional development.

## 5. Conclusions

The theoretical model of social media perception among preservice physical education teachers was explored using Grounded Theory, resulting in the generation of the core category called Social Media Perception System for Preservice Physical Education Teachers. The system contained three subsidiary categories, such as “Overall Perception”, “Value Perception”, and “Risk Perception”. Inside these subsidiary categories, there are different sub-categories inside, like “Overall Perception”, “Media Interface Perception”, “Basic Element Perception”, “Current Status Perception”, and “Development Trend Perception”. Value Perception includes “Information Value Perception”, “Interactive Value Perception”, and “Intelligent Function Perception”. Risk Perception includes “Psychological Risk Perception”, “Information Risk Perception”, and “Privacy Risk Perception”. The above 10 sub-categories and three subsidiary categories are generated based on 70 concepts and 307 labels. This systematic theory has passed the theoretical saturation test and coding consistency test, fully indicating it has good reliability.

The findings of this study on the social media perception among preservice physical education teachers can provide valuable insights into the level of understanding and use of these platforms among this group. Our findings can be used to identify the challenges and concerns that preservice physical education teachers have about using social media for professional development and inform the development of education programs and strategies for addressing these issues. Importantly, this study can help administrative departments to evaluate preservice physical education teachers’ information technology capabilities and determine how to support their use of social media for professional development. Specifically, preservice physical education teachers could benefit greatly from understanding the theme of “Value Perception” in social media. This theme refers to the reasons why they use social media and their expectations for its use, including the informational, communicative, and functional benefits. By comprehending these Value Perceptions, preservice physical education teachers can adopt a more thoughtful and strategic approach to using social media for their professional development. Moreover, by understanding the “Risk Perception” of social media, preservice physical education teachers can be aware of the potential hazards and drawbacks associated with its usage, including psychological, informational, and privacy risks. This awareness can help them to exercise caution in their online behavior and information sharing, as well as take measures to protect their privacy and security. Finally, by grasping their Overall Perception of social media, preservice physical education teachers can be more deliberate and purposeful in their usage, aligning it with their professional development goals while avoiding potential pitfalls.

In order to effectively support the professional development of preservice physical education teachers in the era of information technology, it is important to understand their perceptions of social media and how it can be used for professional development. This study provides valuable insights into the social media perceptions among preservice physical education teachers, but it is recommended that future research be conducted to validate and expand upon these findings. By examining the social media perceptions of preservice physical education teachers in different regions, we can gain a more comprehensive understanding of the challenges and opportunities that these platforms present for professional development in physical education. This knowledge can inform the development of effective strategies for supporting preservice physical education teachers in their use of social media for professional development.

The grounded theory that we used for our study has benefits and also limitations. Grounded theory is a research innovation that holds the potential to greatly benefit the study of constructing Chinese preservice physical education teachers’ perception model of social media in several ways. Firstly, the inductive nature of grounded theory enables the development of a theory that emerges from the data rather than being predetermined by pre-existing theories or models. This approach is particularly advantageous in perception research as it allows the researcher to gain a deeper understanding of the unique perspectives, experiences, and behaviors of the participants in the specific context of Chinese preservice physical education teachers. Furthermore, the iterative process of data collection and analysis inherent in grounded theory enables the refinement of the researcher’s understanding of the participants’ perceptions as more data is collected. This approach results in a more nuanced and comprehensive understanding of the participants’ experiences and perspectives, which is critical in perception research, where the objective is to comprehend how individuals perceive and make sense of their experiences. However, there are also some potential limitations to using grounded theory in this study. One limitation is that grounded theory relies heavily on qualitative data, which can be subjective and open to interpretation. This may result in a limited understanding of the phenomenon being studied if the data is not collected and analyzed rigorously. Moreover, the grounded theory approach requires a significant investment of time and resources, as the researcher must collect, analyze, and interpret a large amount of data to develop a theory that is grounded in the participants’ experiences. This can be a challenge, particularly in cross-cultural research, where there may be language barriers or cultural differences that affect the data collection and analysis process. In conclusion, while grounded theory offers several benefits for the study of Chinese preservice physical education teachers’ perception of social media, there are also potential limitations that must be considered. The researcher must carefully consider these limitations and ensure that the data collection and analysis methods are rigorous and appropriate for the research question being addressed.

## Figures and Tables

**Figure 1 ijerph-20-03886-f001:**
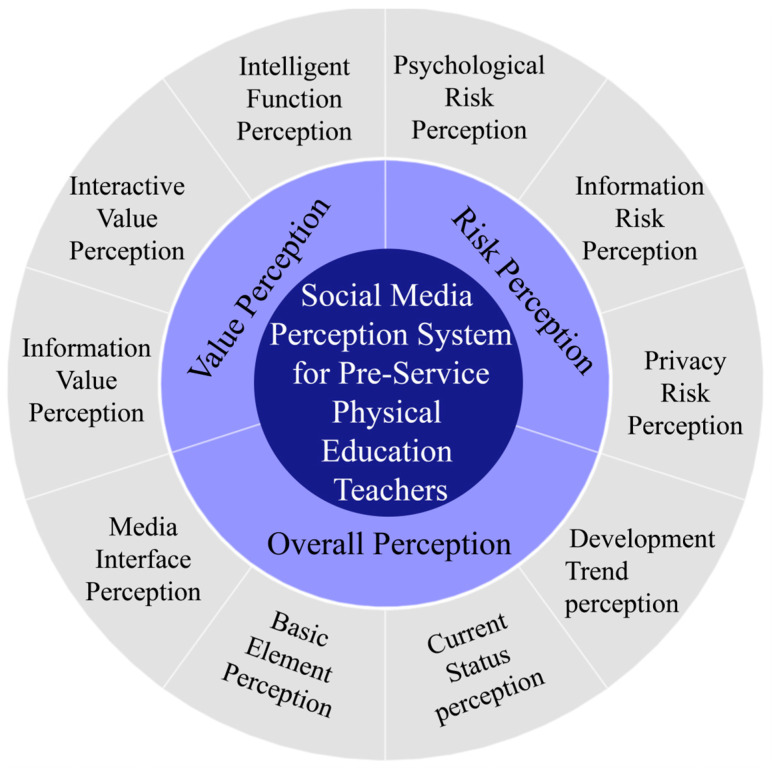
Pre-service physical education teachers’ perception of social media.

**Table 1 ijerph-20-03886-t001:** Demographics for the sample of participants (N = 17).

No.	Gender	Class	Parts	Type of University	Usage Duration (Years)	Usage Frequency (h/per Day)
1	Female	Senior	North-East	Normal College	Five	3–5
2	Male	Junior	South-West	Normal College	Seven	6–8
3	Female	Junior	South-West	Normal College	10	6–8
4	Male	Senior	North-East	Normal College	11	3–5
5	Female	Senior	South-West	Normal College	13	3–5
6	Male	Senior	North-East	Normal College	Eight	3–5
7	Female	Freshman	South-East	Normal College	14	6–8
8	Female	Senior	North-West	Comprehensive University	10	6–8
9	Female	Senior	South-East	Comprehensive University	13	3–5
10	Male	Senior	North-East	Comprehensive University	12	3–5
11	Female	Sophomore	South-East	Comprehensive University	Nine	6–8
12	Male	Junior	South-East	Comprehensive University	Eight	3–5
13	Male	Junior	South-East	Comprehensive University	Seven	9–11
14	Male	Senior	South-West	Comprehensive University	10	6–8
15	Male	Senior	South-East	Sports College	11	3–5
16	Male	Junior	South-East	Sports College	13	6–8
17	Female	Senior	North-East	Sports College	14	9–11

**Table 2 ijerph-20-03886-t002:** Interview Questions.

No.	Interview Questions
1	How do you define the elements of social media and what role you see it playing in your future career as a physical education teacher?
2	What do you think the future trending of using social media in your future job as a physical education teachers? Talk about the ideas and its reasons.
3	What is the advantages and disadvantages of using social media for your professional development in physical education? Talk about it with your social media usage experience.
4	How do you evaluate the credibility of information found on social media in regards to physical education and health?
5	Have you ever encountered false information on social media related to physical education or health? How did you handle the situation?
6	What experience makes you do not want to use social media for your professional development journey as a physical education teacher?

**Table 3 ijerph-20-03886-t003:** Labeling results.

Tagged Labels	Tagged Labels
E1 I feel it is necessary to use social media for motor skill learning	E155 It is difficult to share the physical education knowledge and motor skills learned in class on social media
E2 I find that sports-related learning resources in social media are cutting edge	E156 Unclear how to demonstrate preservice physical education teachers’ professionalism in social media
…….	
E152 Social media is good for maintaining the habit of learning something new every day	E306 It is necessary to ask other users in social media for knowledge and methods of physical and health education
E153 The need to proactively plan application time and learning content in the application of social media	E307 The need to reflect on and master the means of encouraging student communication in social media communication
E154 Social media is very informative about the knowledge needed to plan sports events	

Notes: Reported codes are the conceptual labels applied to the transcribed data. For specific data, see Table A1.

**Table 4 ijerph-20-03886-t004:** Conceptualization results of 307 tagged labels.

Number	Conceptualization Concepts	Tagged Labels	Number of Materials	Number of Participants
D1	I find the graphic videos related to sports or teaching practices on social media smooth, clear and easy to understand	4 articles: E297, E299, E302, E154	54	5
D2	I think there is a lack of communication boards for issues related to physical education	1 Article: E33	3	2
D3	I think pre-service physical education teachers are happy to share or exchange motor knowledge skills in social media	6 articles: E272, E261, E83, E170, E3, E87	21	4
………..				
	Total	307 articles	1122	22

Notes: Articles are the results from the first coding; the participants mean the people referred to the code and the materials mean the interview, online posts and literature referred to the code. For specific data, see Table A2.

**Table 5 ijerph-20-03886-t005:** Results of categorization of 70 concepts.

Subsidiary Categories	Concepts
	D1 I find graphic videos related to sports or teaching practices in social media smooth, clear and easy to understand
C1	D13 I feel that the typographic interface of social media is clean and clear for pre-service physical education teachers
Interface Perception (4)	D55 I think the vivid and colorful interface matches the aesthetics of all pre-service physical education teachers
	D25 I believe that the vehicles of videos, photographs, captions, and others are very suitable for the overall development of pre-service physical education teachers’ professional perceptions
……	
C10	D63 Social media messages from non-sports professionals make me feel very culturally different
Psychological Risk (8)	D38 I feel it is difficult to receive timely responses from high-level physical education teachers or research experts in social media
	D52 Need to have the self-discipline needed to concentrate on learning about sports in social media
	D54 I feel that most of the active sports users are motivated by the intention of selling skills courses or nutritional supplements, etc.
	D69 I feel that the sharing of novel exercise skills in social media is more engaging than in offline classes but it is easy to get distracted

Notes: For specific data, see Table A3.

**Table 6 ijerph-20-03886-t006:** Results of the main axis coding for 10 categories.

Sub-Categories	Subsidiary Categories
B1 Overall Perception	C1 Interface Perception
	C2 Element Perception
	C3 Current Status Perception
	C4 Development Trend Perception
B2 Value Perception	C5 Information Value Perception
	C6 Interactive Value Perception
	C7 Intelligent Function Perception
B3 Risk Perception	C8 Psychological Risk Perception
	C9 Privacy Risk Perception
	C10 Information Risk Perception

## Data Availability

Data can be shared upon reasonable request to the corresponding author.

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
