# Peer review of "Model Construction of Chinese Preservice Physical Education Teachers’ Perception of Social Media: A Grounded Theory Approach"

_ijerph, 2023, doi:10.3390/ijerph20053886_

Round 1

Reviewer 1 Report

Please, see the document attached.

Author Response

Thank you for your valuable feedback, we believe these revisions have strengthened the manuscript and addressed reviewer feedback. Please see the attachment for further details.

Reviewer 2 Report

The development of modern information technologies and the use of social networks create an environment for the exchange of information, interactive communication, learning, and optimization of various processes.

The paper presents a study to explore preservice physical education teachers’ perceptions of social media. According to the authors Given the scholarly attention afforded to social media to promote professional development among physical education teachers, it is necessary to fill the research gap, especially in terms of evidence, population, and methodology.

Strengths:

1.     The presentation was clear and very comprehensible.

2.     In general, the paper presents appropriate organization, and the points advance logically.

3.     Current and pertinent subject matter

4.     The study carried out by the authors presents scientific rigour.

5.     The conclusions obtained are relevant to the scientific area.

6.     The Documentation of sources and references are appropriate and up-to-date

7.     The references are relevant and up-to-date; however, in the content analysis, the: Krippendorff, K., 2004. Content Analysis: An Introduction to its Methodology. London: Sage Publications and L. Bardin. Communication et langages, n°35, 1977. pp. 123-124. are highly cited

8.     The figures and tables are useful.

Weaknesses:

1.     The authors mention that the methodology used was grounded Theory, but I cannot entirely agree. It is impossible based on 17 interviews, with an unsubstantiated selection of candidates. About the sample, we only know some demographic characteristics (Sex, Class, Zone and Kind of university).

“A grounded theory researcher aims to generate a substantive theory that will explain a phenomenon in a specific context and be suited to its intended use. The grounded theory emphasis is theory development (Strauss & Corbin, 1994). Thus, grounded Theory is appropriate when no theory exists or is too abstract to be tested, but it is not appropriate for testing a theory or generating knowledge from objective reality (Martin & Turner, 1986; Suddaby, 2006).

In contrast, a researcher who uses qualitative content analysis aims to "systematically describe the meaning" of materials in a certain respect that the researcher specified from research questions (Schreier, 2012, p. 3).” Although both grounded Theory and qualitative content analysis follow coding processes, the content analysis does not focus on finding relationships among categories or theory building; instead, it focuses on extracting categories from the data.

For the above reasons, the methodology used was an interview survey, and subsequently, the content analysis of the interviews was performed. To proceed with the transcription and content analysis used verbatim ROST CM 6 and NVivo

2.     The research question and objectives should be presented at the beginning of the section. 2. Materials and Methods

3.     The article presents different studies in the area, and there are a large number of researchers publishing in the area under study.

4.     Regarding the interview script, there need to be more indicators to identify the main characteristics of the respondents.

5.     The qualitative analysis also allows for obtaining quantitative data, namely matrix data, word cloud, clusters, etc. The authors did not present any.

Proposals for change:

1.     Rethink the methodology

2.     Improve the literature review, presenting similar studies and talking about the innovation of the study.

3.     In the results, section 3., present quantitative data provided by the NVIVO platform.

4.     Indicate guidelines for future work.

5.     Substantiate with relevant literature citations from the field the results obtained by the authors.

6.     To make the empirical data available in an existing platform for this purpose, for example, the MDPI platform. The article substantially reduces the number of pages, making the data accessible to the community.

Author Response

(The authors gave the same response as above.)

Reviewer 3 Report

Line 30 to 37, you need to considered too about fake news that are growing in the social media.

I think it is important to considered a critical perspective in the use of social media.

About the questions of interview, it is necessary to argument why this questions are general, because the answers can be about personal life and no necessary professional dimension.

Data Analysis: in the abstract was tell that you used atlas ti. and Nvivo, and in the line 152 you said that you used ROST CM 6 software.

Review in table 3.. E4

Organize better categorization, it looks lost after table 4.

Just about curiosity, because in the answer I feel a research more about the social media that the use of it. 

I didn't find they are no worry about physical damage to do exercise in a wrong way, because they can be following a bad influencer, or just do bad a exercise, it wasn't considerate?

Author Response

(The authors gave the same response as above.)

Round 2

Reviewer 2 Report

Revision of the second version

1.    Regarding the methodology adopted, I believe this is a qualitative exploratory study whose research technique is content analysis.
2.    The research question and objectives were included.
3.    The literature review was improved.
4.    The articles present more indicators to identify the main characteristics of the respondents
5.    Regarding quantitative analysis, you can do ( Data exploration: Ask complex questions to discover meaning in your data. Quickly visualise your data with word frequency charts, word clouds, comparison diagrams and more. Look for emerging themes and sentiments using specific queries to identify themes and draw conclusions.
6.    Guidelines for future work have been presented.